# Kinetics of Lignin Separation during the Atmospheric Fractionation of Bagasse with *p*-Toluenesulfonic Acid

**DOI:** 10.3390/ijms23158743

**Published:** 2022-08-06

**Authors:** Baojuan Deng, Yadan Luo, Meijiao Peng, Tao Li, Jianwei Su, Yang Wang, Xuelian Xia, Chengqi Feng, Shuangquan Yao

**Affiliations:** Guangxi Key Laboratory of Clean Pulp & Papermaking and Pollution Control, School of Light Industrial and Food Engineering, Guangxi University, Nanning 530004, China

**Keywords:** *p*-toluenesulfonic, lignin, mechanism, kinetics, shrinking nucleus model

## Abstract

As a green and efficient component separation technology, organic acid pretreatment has been widely studied in biomass refining. In particular, the efficient separation of lignin by *p*-toluenesulfonic acid (*p*-TsOH) pretreatment has been achieved. In this study, the mechanism of the atmospheric separation of bagasse lignin with *p*-TsOH was investigated. The separation kinetics of lignin was analyzed. A non-simple linear relationship was found between the separation yield of lignin and the concentration of *p*-TsOH, the temperature and the stirring speed. The shrinking nucleus model for the separation of lignin was established based on the introduction of mass transfer and diffusion factors. A general model of the total delignification rate was obtained. The results showed that the process of lignin separation occurred into two phases, i.e., a fast stage and a slow stage. The results provide a theoretical basis for the efficient separation of lignin by *p*-TsOH pretreatment.

## 1. Introduction

As inedible plant biomass, lignocellulosic biomass offers great potential for the sustainable production of biofuels and platform chemicals [1,2,3]. However, its stubbornness needs to be overcome. This mainly includes eliminating macroscopic barriers (mass transfer limitations and tissue composition heterogeneity) and microscopic barriers (lignin–carbohydrate crosslinks, etc.) [4]. Fractional separation of lignocellulosic biomass was achieved using different pretreatment techniques [5,6]. Organic acid pretreatment is one of the most competitive methods [7]. The feedstock is depolymerized after organic acid pretreatment. Lignin or hemicellulose is dissolved into a hydrolyzate. Then, a cellulose-rich solid substrate is obtained [8]. It was found that *p*-toluenesulfonic acid (*p*-TsOH) pretreatment has unique advantages. Efficient separation of lignin at atmospheric pressure and low temperature was achieved [9]. High-purity lignin samples were obtained following the efficient cleavage of benzyl ether and benzyl ester bonds. Furthermore, the condensation of lignin was inhibited by *p*-TsOH pretreatment [10]. The enzymatic yield of cellulose was significantly improved [11]. This was attributed to the effective inhibition of the unproductive adsorption of enzymes by lignin [12]. The structural transformation of lignin was analyzed after *p*-TsOH pretreatment [13]. The results indicated that lignin is transformed into a β-O-4-linked α-methoxylated intermediate utilizing *p*-TsOH autocatalysis. Efficient depolymerization and extraction of lignin were achieved. The extracted lignin can be subjected to further modifications and used for various applications. Current research focuses on the principles driving the separation of lignin after *p*-TsOH pretreatment [14]. However, the separation mechanism of lignin, especially, its separation kinetics, after *p*-TsOH pretreatment has not been reported. In fact, kinetic studies of lignin separation can help to further control and optimize *p*-TsOH pretreatment.

So far, the separation kinetics of lignin has been studied in a few cases after organic solvent pretreatment [15,16,17]. The fractionation of Miscanthus and bamboo mixtures using acetic and formic acids was reported by Villaverde [18]. Lignin separation kinetics was analyzed using two parallel first-order reaction models. Shatalov et al. [19] found that lignin polymers can be regarded as individual structural elements or groups of closely related structural elements in a homogeneous lignin separation system. Dagnino et al. [20] treated rice husks with soda–ethanol. The kinetics of lignin extraction was studied. The kinetic constant k_0_ increased from 0.021 min^−1^ to 0.035 min^−1^ at 140–160 °C. The activation energies of the fast and slow phases were 38.59 kJ·mol^−1^ and 33.47 kJ·mol^−1^, respectively. It is believed that the separation of lignin occurs in a quasi-homogeneous system. However, the inhomogeneous distribution of lignin in biomass pellets and the heterogeneous mass transfer and diffusion effects between reactants have been ignored [21]. In fact, the problem of the uneven distribution of lignin was solved by improving the quasi-homogeneous model [22,23]. Reaction parameters describing the potential degree of reaction (solubility of lignin) were introduced. The results showed that the model accuracy was high, and the experimental data were well suited [24]. However, the issue of heterogeneous mass transfer and diffusion between reactants has not been effectively solved. The multilayered structure of the cell wall and the heterogeneity of lignin distribution need to be considered. In fact, an organic acid diffuses from the hydrolyzate through the cell wall to the surface of the unreacted layer during organic acid pretreatment. Then, the reaction of the target product with the organic acid takes place. Degradation products diffuse from the reaction layer into the hydrolyzate. The above organic acid pretreatment involves diffusion and mass transfer processes [21].

Therefore, diffusion and mass transfer effects need to be considered when constructing a kinetic model of lignin separation for organic acid pretreatment. Research found that the shrinking core model is a typical two-phase model [25]; it is often used to study metal dissolution [26], wood autoclave liquefaction [27] and biomass gasification [28] and has revealed the roles of CO_2_ and N_2_ in hydrate formation [29], H_2_ adsorption kinetics [30], solid catalyst catalysis [31] and other phenomena. Zhao et al. [32] modified the classical condensed kernel model. A shrinking core model for the fractionation of bagasse lignin with atmospheric acetic acid was established. The study found that dissolved lignin significantly affects the separation of lignin. Therefore, the reduced core model is a very suitable two-phase model for the separation of lignin by organic acid treatment. It is of great theoretical and practical significance to study the separation kinetics of lignin after *p*-TsOH pretreatment using the shrinking core model.

In this study, the effects of *p*-TsOH concentration, temperature and stirring speed on lignin separation were investigated. A kinetic model for the separation of lignin using *p*-TsOH was established based on the shrinkage core theory. The kinetic equations were solved by comparing the separation coefficient constants. The results provide theoretical support for the study of the reaction mechanism of *p*-TsOH pretreatment in the separation of lignin.

## 2. Results and Discussion

### 2.1. Effects of p-TsOH Pretreatment on the Separation Yield of Lignin

The apparent kinetics of the separation of lignin is influenced by mass transfer processes. Major factors include reaction temperature, pH and degree of mixing [32]. Figure 1 shows the isolated yield of lignin as a function of time at different *p*-TsOH concentrations. The separation of lignin can be divided into two main phases: fast separation and slow separation. First, after 25 min, the separation yield of lignin changed when the concentration of *p*-TsOH was 60% and 65%. In fact, the single yields of lignin increased rapidly to 60.27% and 69.55%, respectively. Then, the rate of lignin separation decreased, and after 60 min, the yields were 60.27% and 74.18%, respectively. This was attributed to residual lignin in the intercellular layer, which is more difficult to remove [8]. It is difficult to obtain an efficient reaction with *p*-TsOH. The isolated yields of lignin rapidly increased to 55.64% and 60.27% in the first 5 min with acid concentrations of 70% and 75%. The growth rate was then slow between 5 and 15 min. However, it increased rapidly between 15 and 25 min and leveled off after 25 min. This was attributed to the increase of *p*-TsOH molecular concentration in solution with the acid concentration. Mass transfer kinetics were enhanced due to the large concentration difference between the two sides of the cell wall. However, the polymerization of dissolved lignin was promoted. *p*-TsOH was consumed in the reaction with condensed lignin and, after the condensed lignin was completely consumed, it continued to react with other unreacted bagasse lignin. In addition, the separation yield of lignin rapidly increased to 60.27% at 80% acid concentration in the first 5 min. Then, it slowly increased to 92.63%, thus remaining almost unchanged. This was attributed to the speed of the fast reaction phase, which was much greater than that of the slow reaction stage. The results showed that both the surface reaction process and the outward diffusion of degradation products from the cell wall were affected by the acid concentration. However, previous studies found that carbohydrates were severely degraded in the presence of 90% *p*-TsOH [33]. Therefore, a concentration of 80% of *p*-TsOH was used to study the separation kinetics of lignin.

As shown in Figure 2, the lignin rapid separation stage was shortened from 25 min to 10 min when the temperature was increased from 50 °C to 60 °C. At the same time, the separation yield of lignin in the rapid separation stage also increased from 32.46% after 25 min to 64.91% after 5 min. This was due to the increase of the reaction rate with the temperature. The time of the slow delignification stage of *p*-TsOH gradually shortened with an increasing temperature. The isolated yield of lignin gradually increased with time at 80 °C and 90 °C. However, a second rapid increase was found after 25 min. Finally, the separation yields of lignin were 88.09% and 90.14%, respectively. This was attributed to the condensation of lignin and the attachment of carbohydrates to lignin [34], which became the main controlling step in the later stage of the lignin separation reaction. However, polycondensation of lignin is inevitable at high temperatures and results in a suppressed rate of lignin separation [35].

The effect of agitation rate on the yield of lignin separation was investigated in the *p*-TsOH treatment process shown in Figure 3. The reaction proceeded slowly at zero stirring speed. The isolated yield of lignin was 12.52% in 30 min. Then, the variation of the lignin separation yields flattened. It was 17.58% at 120 min. The overall delignification was maintained at a low level. This is attributed to the uneven mixing of bagasse and *p*-TsOH solution. The contact of H_3_O^+^ in solution with unreacted moieties is hindered [33]. Obviously, the degree of separation of lignin was improved by increasing the stirring speed. The separation yield of lignin was 54.50% at 15 min when the stirring speed was increased to 300 rpm. It increased to 72.64% at 120 min. However, a significant improvement in the separation yield of lignin is difficult to achieve by continuing to increase the rotational speed. The lignin separation yield was almost equal to 300 rpm when the stirring speed reached 900 rpm. Therefore, uniform mixing of the reactants was achieved when the stirring speed reached 300 rpm. The isolated yield of lignin remained unchanged by further increasing the stirring speed. This means that a dynamic equilibrium of the reaction is obtained.

In summary, the separation of lignin can be divided into two distinct processes: a stage where lignin is rapidly separated at the beginning of the reaction and a stage where lignin is slowly separated towards the end of the reaction. Therefore, finding a suitable kinetic model and establishing a kinetic model for the interpretation of these two stages is crucial.

### 2.2. Model Derivation for the Separation Kinetics of Lignin

Plant cell walls are usually structurally complex, as shown in Figure 4a. From the outside to the inside, there are, respectively, a primary wall and a secondary wall. The secondary wall further comprises an outer layer, a middle layer and an inner layer, of which the middle layer is the thickest [36]. The process of separating lignin from *p*-TsOH involves the transfer of *p*-TsOH from the intercellular layer to the primary wall and finally to the secondary wall, while dissolved lignin moves from the secondary wall to the primary wall and finally dissolves from the intercellular layer. Because the lignin and hemicellulose of the cell wall are dissolved in the reaction cycle, the unreacted layer is gradually compressed. Therefore, the separation of lignin by *p*-TsOH fits the reduced core model [25]. In this study, the slab shrinkage model proposed by Zhao et al. [32] was considered to better understand the *p*-TsOH-based delignification process. The bagasse cell wall can be simplified into a flat plate. The details are shown in Figure 4b.

The following *p*-TsOH separation process of bagasse lignin is proposed. It includes several steps:

The first step is the outward diffusion of *p*-TsOH. *p*-TsOH molecules diffused from the *p*-TsOH solution, through the boundary layer (2), to the outer surface of the reaction layer (3). *p*-TsOH concentration decreased, from the concentration C_p1_ in the liquid phase to the concentration C_p2_ on the surface of the reaction layer (3).

The second step is the internal diffusion of *p*-TsOH. *p*-TsOH molecules diffused from the solid surface into the unreacted layer (4) through the reactive layer (3). *p*-TsOH concentration decreased to C_p3_.

The third step is the surface reaction of lignin separation. The reaction was carried out on the surface of the unreacted layer (4). It mainly included the reaction between lignin and hemicellulose in the cell wall (the degradation of lignin is mainly discussed here) and *p*-TsOH molecules. The unreacted layer (4) gradually shrank as the reaction proceeded. At this time, the concentration of lignin dissolved on the surface of the unreacted layer was C_L3_.

The fourth step is the internal diffusion of lignin. The portion of lignin dissolved on the surface of the unreacted layer (4) diffused to the surface of the solid particles through the reaction layer (3). Its concentration decreased from C_L3_ to C_L2_.

The fifth step is the out-diffusion of lignin. Lignin further diffused into the liquid phase through the boundary layer (2). Its concentrations ranged from C_L2_ to C_L1_.

The liquid–solid reaction during the separation of lignin by *p*-TsOH is described by Equation (1).
P(l) + B(s) → F(l) + R(s)(1)

In the formula, P is a *p*-TsOH molecule, B is lignin, F is the liquid-phase degradation product (dissolved lignin), R represents other unreacted components in the cell wall, and l and s denote the liquid and solid phases, respectively.

The reaction is believed to be carried out in an isothermal environment. Therefore, the effects of temperature on the diffusion and reaction rate constants are not directly considered in the model. A mass balance-based plate shrinkage model was established assuming that the cell wall is a square with side length of 1.

The external diffusion rate of *p*-TsOH is obtained by Equation (2).
(2)$$-r_{p_{2}}=k_{p_{2}}\;\left(C_{p_{1}}-C_{p_{2}}\right)$$
where r_p2_ is the extramolecular diffusion rate of *p*-TsOH, k_p2_ is the diffusion coefficient of the boundary film for outward diffusion on the solid surface, and C_p1_ and C_p2_ are *p*-TsOH molecular concentrations on the liquid and solid surfaces, respectively.

The internal diffusivity is expressed by Equation (3).
(3)$$-r_{p_{3}}=k_{p_{3}}{\left(\frac{{\mathrm{dC}}_{p1}}{\mathrm{dL}}\right)}_{L_{C}}=K_{p_{3}}C_{p_{3}}^{\alpha }C_{L_{3}}^{\beta }\;$$
where r_p3_ is the intramolecular diffusion rate of *p*-TsOH, and k_p3_ is the effective diffusion coefficient of *p*-TsOH molecules in the reaction layer. Additionally, interactions exist between *p*-TsOH molecules and dissolved lignin. Therefore, the *p*-TsOH molecules reacting with dissolved lignin have the same diffusion coefficient as that of dissolved lignin; r_p3_ is the internal diffusion rate, K_p3_ is the internal diffusion apparent rate constant, α is the reaction order of the concentration of *p*-TsOH molecules, and C_p3_ is the concentration of *p*-TsOH molecules on the surface of the unreacted layer. β is the reaction order of lignin concentration, and C_L3_ is the concentration of lignin on the surface of the unreacted layer.

At steady state, no net *p*-TsOH accumulates, so Equation (4) is obtained:(4)$${\left(k_{p_{3}}\frac{{\mathrm{dC}}_{p_{1}}}{\mathrm{dL}}\right)}_{L}={\left(k_{p_{3}}\frac{{\mathrm{dC}}_{p_{1}}}{\mathrm{dL}}\right)}_{L+\mathrm{dL}}$$

Considering the boundary conditions, C_p1_(L_S_) = C_p2_; and C_p1_(L_C_) = C_p3_, and integrating Equation (4) twice we obtain:(5)$$C_{p_{1}}-C_{p_{3}}=\left(C_{p_{2}}-C_{p_{3}}\right)\frac{L-L_{C}}{L_{S}-L_{C}}$$
and
(6)$${\left(\frac{{\mathrm{dC}}_{p_{1}}}{\mathrm{dL}}\right)}_{L_{C}}=\frac{\left(C_{p_{2}}-C_{p_{3}}\right)}{L_{S}-L_{C}}$$

Therefore, when introducing (6) into (3):(7)$$-r_{p}=k_{p_{3}}\frac{\left(C_{p_{2}}-C_{p_{3}}\right)}{L_{S}-L_{C}}$$

Combining Equations (2), (5) and (7) to eliminate C_p2_ and C_p3_, we obtain:(8)$$-r_{p}=k_{p_{3}}\frac{k_{p_{1}}C_{p_{1}}^{\alpha }}{\left(\frac{k_{p_{1}}}{k_{p_{2}}}+\frac{k_{p_{1}}\left(L_{S}-L_{C}\right)}{k_{p_{3}}}+1\right)}$$

The molar amount of dissolved lignin after separation is:(9)$$n_{L}=\frac{\rho_{L}V}{M_{L}}=\frac{\rho_{L}}{M_{L}}L_{C}$$
where n_L_ is the molar amount of dissolved lignin, ρ_L_ is the mass density of lignin in the cell wall, that is, the content of lignin in the cell wall, M_L_ is the average molecular weight of lignin. Equation (10) is obtained:(10)$$-r_{L}=-\frac{{\mathrm{dn}}_{L}}{\mathrm{dt}}=\frac{\rho_{L}}{M_{L}}\frac{{\mathrm{dL}}_{C}}{\mathrm{dt}}$$
where r_L_ is the reaction rate of lignin separation.

Substituting Equation (10) into Equation (8), the differential relationship between the unreacted layer thickness L_C_ and the time t can be expressed as:(11)$$-\frac{{\mathrm{dL}}_{C}}{\mathrm{dt}}=\frac{k_{p_{1}}M_{L}C_{p_{1}}^{\alpha }}{\rho_{L}{\left(\;\frac{k}{k_{p_{2}}}+\frac{k\left(L_{S}-L_{C}\right)}{k_{p_{3}}}+1\right)}^{\alpha }}$$

Since cell wall thickness is mainly due to the secondary wall middle layer [36], and lignin is usually evenly distributed in the secondary wall middle layer, the assumption that lignin is distributed in the cell wall is valid. The relationship between the degree of separation of lignin (X_L_) and the thickness of the unreacted layer L_C_ can be expressed as:(12)$$X_{L}=\frac{\left(L_{S}-L_{C}\right)\rho_{L}}{L_{S}\rho_{L}}=1-\frac{L_{C}}{L_{S}}$$

When the surface reaction of the unreacted layer in the separation of lignin is a first-order reaction with respect to *p*-TsOH concentration, that is, when α = 1, the integral Equation (12) is:(13)$$-\int_{L_{S}}^{L_{S}}(\frac{1}{k_{p_{2}}}+\frac{L_{S}-L_{C}}{k_{p_{3}}}+\frac{1}{k_{p_{1}}}){\mathrm{dL}}_{C}=\frac{M_{L}C_{p_{1}}}{\rho_{L}}\int_{0}^{t}\mathrm{dt}$$

The result is as follows:(14)$$\left(\frac{1}{k_{p_{1}}}+\frac{1}{k_{p_{2}}}+\frac{L_{S}}{k_{p_{3}}}\right)\left(L_{S}-L_{C}\right)+\frac{L_{C}^{2}-L_{S}^{2}}{2k_{p_{3}}}=\frac{M_{L}C_{p_{1}}}{\rho_{L}}t$$

The relationship between the separation degree of lignin (X_L_) and time (t) is as follows:(15)$$X_{L}\left(\frac{1}{k_{p_{1}}}+\frac{1}{k_{p_{2}}}\right)L_{S}+\frac{L_{S}^{2}X_{L}}{2k_{p_{3}}}=\frac{M_{L}C_{p_{1}}}{\rho_{L}}t$$

Therefore, Equation (15) is a general model of the rate of lignin separation during the first-order surface reaction involving *p*-TsOH.

In the early stage of the reaction, lignin separation is rapid. When internal diffusion is the control step, that is, when k_p3_ ≪ k_p2_ and k_p1_, C_p1_ ≈ C_p2_, C_p2_ ≫ C_p3_, Equation (15) can be expressed as:(16)$$X_{L}=\frac{1}{L_{S}}\sqrt{\frac{2k_{p_{3}}M_{L}C_{p_{1}}}{\rho_{L}}t}$$

In the later stage of the reaction, lignin separation is slow, and when the surface reaction is the limit step, that is, when k_p1_ ≪ k_p2_ and k_p3_, C_p1_≈C_p2_≈C_p3_, Equation (15) can be expressed as:(17)$$X_{L}=\frac{k_{p_{1}}M_{L}C_{p_{1}}}{\rho_{L}L_{S}}t$$

### 2.3. Analysis of the Separation Kinetic Equation of Lignin

The results show that the process of separation of bagasse lignin by *p*-TsOH pretreatment can be described using a flat-plate shrinkage nucleation model. According to this model, the separation of lignin is carried out layer by layer. In fact, the absence of cellulose in the reaction was found in a previous study [13]. This means that the reaction of organic acids with cellulose can be ignored. The fitting results of lignin separation kinetics are shown in Figure 5. It was found that the separation rate of lignin increased nonlinearly with the increase of time (t). This was attributed to the fact that the separation rate is controlled synergistically by the internal diffusion of lignin and the surface reaction.

The fitting diagrams of the separation degree and time of *p*-TsOH-pretreated lignin at different temperatures are shown in Figure 5a. The molecular motion of *p*-TsOH molecules was slower when the temperature was below 50 °C. Therefore, the separation yield of lignin showed a slow increase. The whole kinetic process involved a fast delignification stage and a slow delignification stage. The separation degree (X_L_) of lignin as a function of time (t) was obtained by introducing the experimental data into the initial stage of the reaction to fit Equation (16). The results are shown in Figure 5a. Table 1 shows that the lignin separation coefficient increased with increasing temperature. It increased from 9.046 × 10^−11^ L∙mol^−1^∙s^−1^ to 22.206 × 10^−11^ L∙mol^−1^∙s^−1^ when the temperature increased from 60 °C to 80 °C and *p*-TsOH concentration was 80%. Then, a linear relationship between the degree of separation of lignin and time (t) was obtained, fitting the experimental data into Equation (17) at the later stage of the reaction. However, the lignin separation coefficient was 10^−13^ L∙mol^−1^∙s^−1^. The separation efficiency of the fast lignin separation stage was two orders of magnitude higher than that of the slow lignin separation stage. In particular, the separation efficiency of the fast lignin separation stage could be three orders of magnitude higher than that of the slow lignin separation stage at the optimum temperature. Therefore, a rapid diffusion of lignin in the early stage is achieved by increasing the temperature.

The fitting of *p*-TsOH concentration to the separation degree and time of lignin is shown in Figure 5b. It was found that the separation coefficient of lignin increased with the increase of *p*-TsOH concentration in the early stage of the reaction. It was 4.936 × 10^−11^ L∙mol^−1^∙s^−1^ when *p*-TsOH concentration was 70%. It improved by one order of magnitude at 80% *p*-TsOH concentration. This indicates that the concentration of *p*-TsOH has a significant effect on the separation rate of lignin. However, the separation coefficient of lignin first increased and then decreased with the increase of *p*-TsOH concentration at the later stage of the reaction. This is attributed to the fact that *p*-TsOH molecules in solution reacted with the dissolved lignin at high acid concentrations. Difficult-to-dissolve macromolecules form from the condensation reaction of lignin [37]. The fitting R^2^ of the kinetic data was within 10% in both the fast separation of lignin and the slow separation of lignin. This shows that the fitting effect was good. Additionally, in previous work [33], we determined that the separation of lignin by *p*-TsOH was best at acid concentrations of 80% and at 80 °C. In this experiment, the acid concentration obtained with *p*-TsOH was 80% and 80 °C, and the lignin separation coefficient in the stage of rapid separation was 2.2206 × 10^−10^ L∙mol^−1^∙s^−1^, and R^2^ was 0.9130, The lignin separation coefficient in the slow delignification stage was 1.2836 × 10^−13^ L∙mol^−1^∙s^−1^, and R^2^ was 0.9782. Therefore, the rapid diffusion of lignin in the early stage was enhanced by increasing acid concentration and temperature, and the surface reaction became the controlling step due to the depolymerization and condensation of the dissolved lignin in the later stage of the reaction, so the lignin separation coefficient decreases. The estimates of the models employed in this work are consistent with previous findings.

## 3. Materials and Methods

### 3.1. Materials

Bagasse was obtained from a local sugar factory (Guangxi, China). It was dried, screened and crushed. We obtained 60-mesh bagasse powders. The chemical composition was analyzed using National Renewable Energy Laboratory methods. The specific steps and methods are described by Ge and collaborators [3]. The contents of cellulose, hemicellulose and lignin were 46.69%, 19.81% and 21.33%, respectively. *p*-toluenesulfonic acid was purchased from Sigma-Aldrich (Shanghai, China). Other analytical chemicals were purchased from Aladdin (Shanghai, China).

### 3.2. Isolation of Lignin

The *p*-toluenesulfonic acid pretreatment was carried out in a reactor (BS500, Laibei, Shanghai, China) s equipped with a magnetic stirrer, a heater and a temperature controller. We added 20 g of bagasse powder to a *p*-TsOH solution (1:20 solid/liquid ratio). The separation reaction of lignin was carried out at different acid concentrations, temperatures, stirring revolutions and sampling times. The reaction was quickly stopped by adding 0 °C deionized water until the hydrolyzate was diluted to neutrality. Lignin samples were obtained by centrifugation, washing and freeze-drying. The kinetic experimental data were obtained by varying *p*-TsOH concentration, reaction temperature and rotational speed. The concentration of *p*-TsOH was 60–80%, the reaction temperature was 50–90 °C, and the rotation speed was 0–900 rpm. The sampling time was 2 min, 5 min, 10 min, 15 min, 25 min, 40 min and 60 min.

### 3.3. Content and Properties of the Isolated Lignin

The acid-soluble lignin content in the hydrolyzate was analyzed using a UV–Vis spectrophotometer (Agilent 8453, Agilent, Santa Clara, CA, USA). Acid-insoluble lignin was measured gravimetrically. The isolated yield of lignin was calculated [38]. The molecular weight of the isolated lignin was analyzed using gel permeation chromatography (GPC). The molecular weight of the acetylated lignin samples was determined by HPLC (Agilent 1260 Infinity II, Agilent, CA, USA). Using tetrahydrofuran as the eluent, the column temperature was 30 °C; lignin content was determined at 278 nm by a UV detector (using the peak position of polystyrene as the standard) [39].

## 4. Conclusions

The effects of *p*-TsOH concentration, reaction temperature and stirring speed on lignin separation were investigated. A kinetic model of lignin separation using *p*-TsOH was established based on the “plate shrinkage model”. The results showed that the separation of lignin during the reaction clearly involved two stages. The formula indicating the degree of separation of lignin at the initial stage of the reaction is $X_{L}=\frac{1}{L_{S}}\sqrt{\frac{2k_{p_{3}}M_{L}C_{p_{1}}}{\rho_{L}}t}$; the formula indicating the degree of separation of lignin at the end of the reaction is $X_{L}=\frac{k_{p_{1}}M_{L}C_{p_{1}}}{\rho_{L}L_{S}}t$. The optimal lignin separation kinetic parameters regarding *p*-TsOH were compared, and the fitting difference of the lignin separation coefficient between the two stages was within 10%. Therefore, the lignin separation model we propose provides a theoretical reference for the organic acid-based separation of lignin and the pretreatment with an organic acid for the fractionation of lignocellulose.

## Figures and Tables

**Figure 1 ijms-23-08743-f001:**
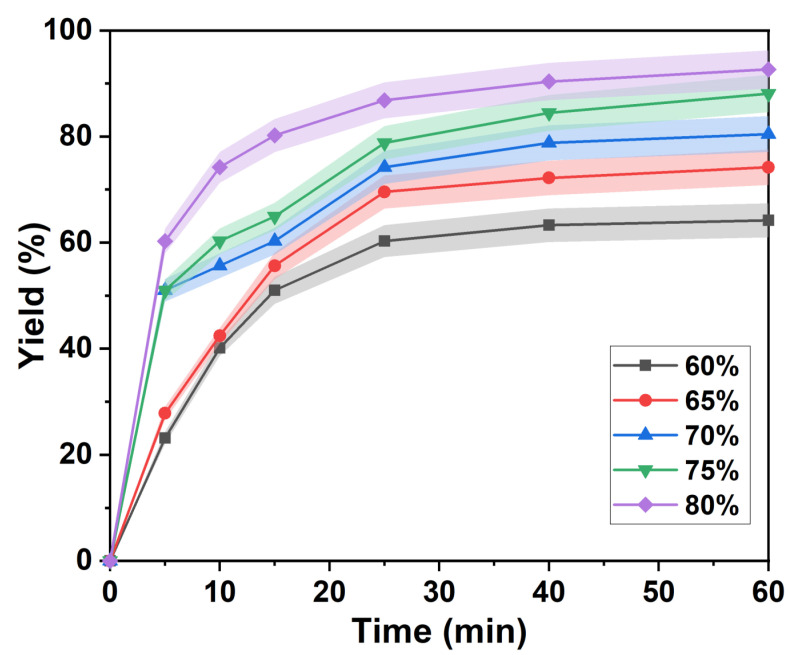
Lignin separation yield in the presence of different *p*-TsOH concentrations.

**Figure 2 ijms-23-08743-f002:**
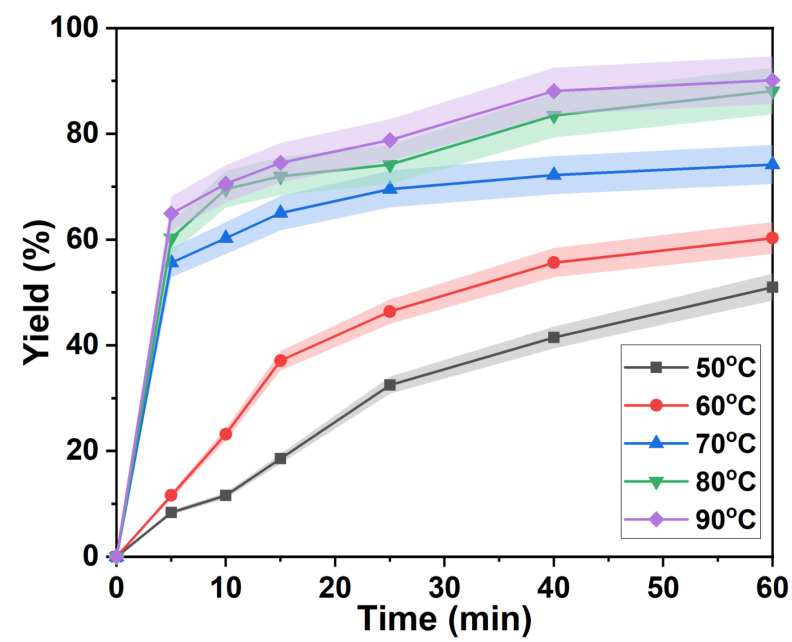
Lignin separation yield at different temperatures.

**Figure 3 ijms-23-08743-f003:**
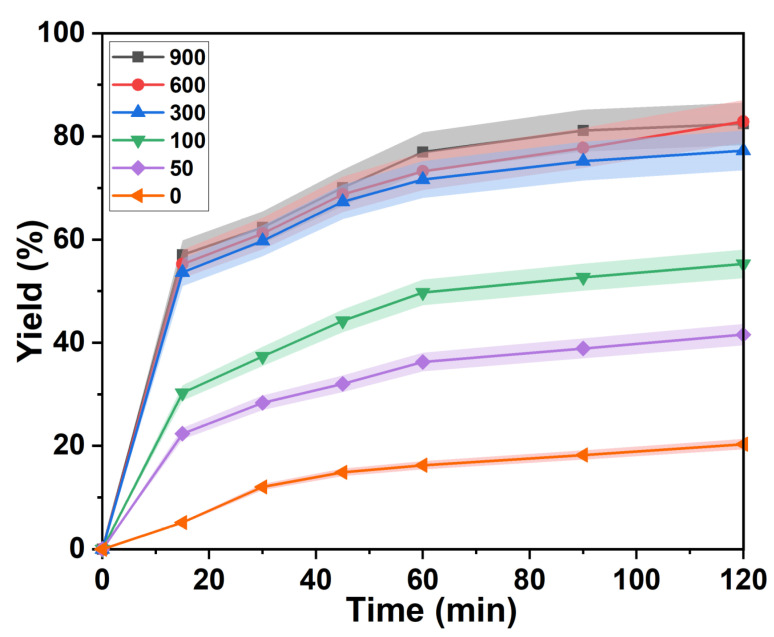
Change of lignin separation yield under different stirring speed.

**Figure 4 ijms-23-08743-f004:**
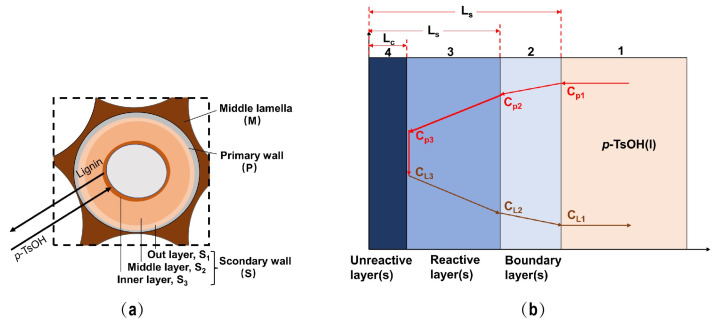
(**a**) Separation of lignin by *p*-TsOH pretreatment; (**b**) Schematic diagram of the plate core shrinking model.

**Figure 5 ijms-23-08743-f005:**
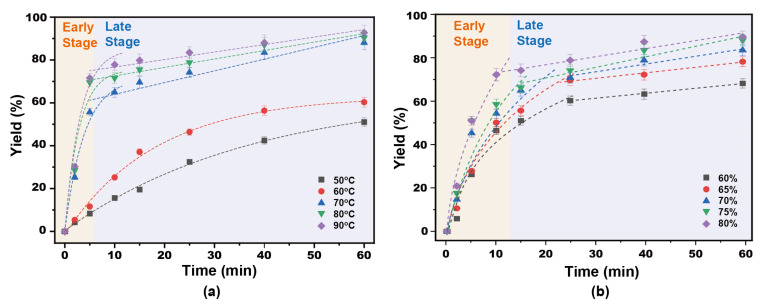
Kinetic fitting curve of lignin separation following *p*-TsOH treatment: (**a**) Influence of reaction temperature; (**b**) Influence of *p*-TsOH concentration.

**Table 1 ijms-23-08743-t001:** Determination of the model parameters for the fast delignification stage and the slow delignification stage based on the experimental data.

Concentration (%)	Temperature (°C)	Rapid Separation of Lignin Stage		Slow Separation of Lignin Stage	
		K (10^−11^ L mol^−1^ s^−1^)	R^2^	K (10^−13^ L mol^−1^ s^−1^)	R^2^
70	60	3.3120	0.9002	8.6335	0.9012
70	70	4.5910	0.8944	8.4343	0.8897
70	80	4.9363	0.9018	7.6945	0.9623
75	60	5.0220	0.9021	8.6833	0.9461
75	70	5.3572	0.8945	8.8770	0.8966
75	80	7.0500	0.9758	9.8557	0.9879
80	60	9.0461	0.8945	8.6152	0.9328
80	70	16.4930	0.8935	6.2514	0.9638
80	80	22.2060	0.9130	1.2836	0.9782

## Data Availability

The data presented in this study are available in the manuscript’s figure.

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
