# Peer review of "Kinetics of Lignin Separation during the Atmospheric Fractionation of Bagasse with p-Toluenesulfonic Acid"

_ijms, 2022, doi:10.3390/ijms23158743_

Round 1
Reviewer 1 Report
The study contributes kinetics of lignin separation scheme for organic acid pretreatment, which provides a theoretical basis for the efficient separation of lignin by p-toluenesulfonic acid pretreatment. In addition, the influence of concentration of p-toluenesulfonic acid, temperature and stirring rate on the separation rate was also investigated. In short, this manuscript is well written. I think it could be published with some revisions.
1, In line 278, “the lignin separation coefficient was only 10-13 L∙mol-1∙s-1”, but in the table 1, the units of K in the stage of slow separation of lignin is “10-13 L∙mol-1∙s-1”, So may “10-13 L∙mol-1∙s-1” should be “10-13 L∙mol-1∙s-1”. Check these date carefully.
2, In lines 325-326, “The reaction was quickly stopped by adding 0 oC deionized water. The hydrolyzate was diluted to neutrality.” Are these two sentences related? What amount of 0 oC deionized water is required to stop the reaction should be described in detail.
3, In line 339, “oC” should be changed to oC. Please check date carefully again.
4, Please check the references carefully for formatting problems. In addition, do you have conducted the composition analysis of the residue during the delignification process? This data can better reflect the delignification process, and the yield is generally lower than the delignification rate. Therefore, if this data is provided, I think the experimental results are more reasonable.
5, Why do you use the “shrinking nucleus model” in this study? What advantages do you think this model has for explaining the mechanism of delignification process?
6, Why only the dissolution of lignin was considered in the model hypothesis and whether the dissolution of cellulose and hemicelluloses had an effect on the dissolution of lignin?
Author Response
Dear Reviewer,
Thank you for your letter and for the comments concerning our manuscript entitled “Kinetics of lignin separation during atmospheric fractionation of bagasse with p-toluenesulfonic acid”. We have studied your comments carefully and have made corrections which we hope could meet your requirements. All changes were marked up using the “Track Changes” function.
Questions you put forward are explained as follows:
1. In line 278, “the lignin separation coefficient was only 10-13 L∙mol-1∙s-1”, but in the table 1, the units of K in the stage of slow separation of lignin is “10-13 L∙mol-1∙s-1”, So may “10-13 L∙mol-1∙s-1” should be “10-13 L∙mol-1∙s-1”. Check these date carefully.
It was modified according to the comments.
2. In lines 325-326, “The reaction was quickly stopped by adding 0 oC deionized water. The hydrolyzate was diluted to neutrality.” Are these two sentences related? What amount of 0 oC deionized water is required to stop the reaction should be described in detail.
Thanks to the reviewer for the question. “The reaction was quickly stopped by adding 0oC deionized water” until “the hydrolyzate was diluted to neutrality”. Because the purpose of adding hydrolysate to cold water is rapid cooling, and excessive cold water will affect the concentration determination of the main components of hydrolysate. Therefore, the hydrolysate should be diluted to neutral. It was modified according to the comments.
3. In line 339, “oC” should be changed to oC. Please check date carefully again.
It was modified according to the comments.
4. Please check the references carefully for formatting problems. In addition, do you have conducted the composition analysis of the residue during the delignification process? This data can better reflect the delignification process, and the yield is generally lower than the delignification rate. Therefore, if this data is provided, I think the experimental results are more reasonable.
Thank you for your questions, the references carefully for formatting problems was modified according to the comments. The composition analysis of the residue is not showed here, because we have discussed the compositional changes of cellulose, hemicellulose and lignin in the residue in detail in article Acidolysis mechanism of lignin from bagasse during p-toluenesulfonic acid treatment (Feng, C.; Zhu, J.; Cao, L.; Yan, L.; Qin, C.; Liang, C.; Yao, S. Acidolysis mechanism of lignin from bagasse during p-toluenesulfonic acid treatment. Ind Crop Prod 2022, 176, 114374). In this paper, the kinetic changes of lignin are mainly studied, so no longer repeat.
5. Why do you use the “shrinking nucleus model” in this study? What advantages do you think this model has for explaining the mechanism of delignification process?
Thanks to the reviewer for the question. The separation process of lignin by p-TsOH is that p-TsOH goes from the intercellular layer to the primary wall and finally to the secondary wall, while the dissolved lignin is from the secondary wall to the primary wall, and finally to the intercellular layer. During the whole process, the unreacted layer is gradually reduced and compressed, which meets the requirements of the “shrinking nucleus model”. The biggest advantage of this model is that it takes into account the inhomogeneous distribution of lignin in the cell wall and the diffusion between the liquid and solid phases.
6. Why only the dissolution of lignin was considered in the model hypothesis and whether the dissolution of cellulose and hemicelluloses had an effect on the dissolution of lignin?
In this model, according to our previous studies (Feng, C.; Zhu, J.; Cao, L.; Yan, L.; Qin, C.; Liang, C.; Yao, S. Acidolysis mechanism of lignin from bagasse during p-toluenesulfonic acid treatment. Ind Crop Prod 2022, 176, 114374), the dissolution of cellulose can be treated as an “inert component” compared with the dissolution of lignin, while the dissolution mechanism of hemicellulose has been studied in previous studies (Feng, C.; Zhu, J.; Hou, Y.; Qin, C.; Chen, W.; Nong, Y.; Liao, Z.; Liang, C.; Bian, H.; Yao, S. Effect of temperature on simultaneous separation and extraction of hemicellulose using p-toluenesulfonic acid treatment at atmospheric pressure. Bioresour. Technol. 2022, 348, 126793). In addition, we believe that hemicellulose dissolution in the study of lignin is a complex content, and the model and method will need to be improved, which is the focus of our next study.
Reviewer 2 Report
After reading the manuscript, I regard it a good paper worthy of publication in this journal. The experimental part seems to be well carried out and their findings are interesting. My only comment is related to the model. I do not understand equation 3 since it shows the same constant kp3 in both sides and that cannot be. The term on the left corresponds to Fick Law and kp3 is an effective diffusion coefficient but the term on the right is a chemical reaction and it must display an appropiate kinetic constant, not the same kp3. The authors should clarify this point before acceptance.
Author Response
Dear Reviewer,
Thank you for your letter and for the comments concerning our manuscript entitled “Kinetics of lignin separation during atmospheric fractionation of bagasse with p-toluenesulfonic acid”. We have studied your comments carefully and have made corrections which we hope could meet your requirements. All changes were marked up using the “Track Changes” function.
Questions you put forward are explained as follows:
After reading the manuscript, I regard it a good paper worthy of publication in this journal. The experimental part seems to be well carried out and their findings are interesting. My only comment is related to the model. I do not understand equation 3 since it shows the same constant kp3 in both sides and that cannot be. The term on the left corresponds to Fick Law and kp3 is an effective diffusion coefficient but the term on the right is a chemical reaction and it must display an appropiate kinetic constant, not the same kp3. The authors should clarify this point before acceptance.
Thanks to the reviewer for patiently pointing out the error. kp3 on the right side of equation (3) should be marked as Kp3. In order not to affect subsequent understanding, all the formulas in the article have been corrected.